# Compositional Abilities Emerge Multiplicatively: Exploring Diffusion Models on a Synthetic Task

**Maya Okawa**[*1,2], **Ekdeep Singh Lubana**[*1,3], **Robert P. Dick**[3], **Hidenori Tanaka**[*1,2]

[1]Center for Brain Science, Harvard University, Cambridge, MA, USA
[2]Physics & Informatics Laboratories, NTT Research, Inc., Sunnyvale, CA, USA
[3]EECS Department, University of Michigan, Ann Arbor, MI, USA

## Abstract

Modern generative models exhibit unprecedented capabilities to generate extremely realistic data. However, given the inherent compositionality of the real world, reliable use of these models in practical applications requires that they exhibit the capability to compose a novel set of concepts to generate outputs not seen in the training data set. Prior work demonstrates that recent diffusion models do exhibit intriguing compositional generalization abilities, but also fail unpredictably. Motivated by this, we perform a controlled study for understanding compositional generalization in conditional diffusion models in a synthetic setting, varying different attributes of the training data and measuring the model's ability to generate samples out-of-distribution. Our results show: (i) the order in which the ability to generate samples from a concept and compose them emerges is governed by the structure of the underlying data-generating process; (ii) performance on compositional tasks exhibits a sudden "emergence" due to multiplicative reliance on the performance of constituent tasks, partially explaining emergent phenomena seen in generative models; and (iii) composing concepts with lower frequency in the training data to generate out-of-distribution samples requires considerably more optimization steps compared to generating in-distribution samples. Overall, our study lays a foundation for understanding emergent capabilities and compositionality in generative models from a data-centric perspective.

## 1 Introduction

The scaling of data, models, and computation has unleashed powerful capabilities in generative models, enabling controllable synthesis of realistic images [1–11], 3D scenes [12–16], videos [17–25], accurate image-editing [26–31], and semantically coherent text generation [32–46]. With increased interest in incorporating these models in day-to-day applications [47–50], e.g., to improve robotic systems via better planning and grounding [51–59], analyzing and improving their reliability has become crucial.

With the motivation above, we study compositional generalization abilities of conditional diffusion models, i.e., diffusion models that are conditioned on auxiliary inputs to control their generated images (e.g., text-conditioned diffusion models [6, 26]). Given the inherent compositionality of the real world, it is difficult to create a training dataset that exposes a model to all combinations of different concepts underlying the data-generating process. We therefore argue that compositional generalization is central to model reliability in out-of-distribution scenarios, i.e., when the model has to compose a novel set of concepts to generate outputs not seen in the training data set [60, 61]. As shown by several prior works investigating the compositional generalization capabilities of off-the-shelf text-conditioned diffusion models [62–72], modern diffusion models often compose complicated

---

*Equal contribution. Email: {mayaokawa, ekdeeplubana, hidenori_tanaka}@fas.harvard.edu.

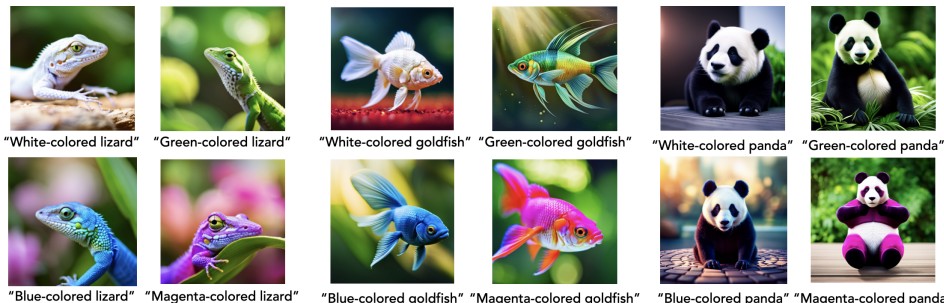

"White-colored lizard"  "Green-colored lizard"    "White-colored goldfish"  "Green-colored goldfish"    "White-colored panda"  "Green-colored panda"

"Blue-colored lizard"  "Magenta-colored lizard"    "Blue-colored goldfish"  "Magenta-colored goldfish"    "Blue-colored panda"  "Magenta-colored panda"

Figure 1: **(Lack of) Compositionality in text-conditioned diffusion models.** Images generated using Stable Diffusion v2.1 [73]. We see diffusion models conditioned on text descriptions of the concepts in an image often allow generation of novel concepts that are absent from the training data, indicating an ability to compose learned concepts and generalize out-of-distribution (lizard and goldfish panels). However, the model sometimes unpredictably fails to compose concepts it has learned, i.e., compositional capabilities depend on the specific concepts being composed. For ex., generating a panda in the above figure is difficult for the model, likely because it has not been exposed to different colors of pandas. The model instead alters the background or lighting to change the color.

concepts, producing entirely non-existent objects, but also unpredictably fail to compose apparently similarly complicated concepts (see Fig. 1). It remains unclear why models produce novel samples by composing some concepts but not others. Indeed, what concepts does a model learn to compose? What differentiates these concepts from the ones the model is unable to compose?

**"Model Experimental Systems" Approach with Interpretable Synthetic Data.** To address the above questions, we seek to take a systematic approach to understanding complex systems, such as generative models, by dividing the process into hypothesis generation and testing. Specifically, drawing inspiration from the natural sciences, we adopt a *model-experimental systems approach*, which involves studying simplified, controllable, and steerable experimental systems to develop mechanistic hypotheses applicable to more complex systems. For instance, in neuroscience research, model organisms like mice or fruit flies are used to study neural mechanisms and develop hypotheses that can be tested on the larger-scale human brain [74–76]. Similarly, in our work, we design a synthetic experimental setup that pursues simplicity and controllability while preserving the essence of the phenomenon of interest, i.e., compositional generalization. Specifically, our data-generating process tries to mimic the structure of the training data used in text-conditioned diffusion models by developing pairs of images representing geometric objects and tuples that denote which concepts are involved in the formation of a given image (see Fig. 2). We train diffusion models on synthetic datasets sampled from this data-generating process,

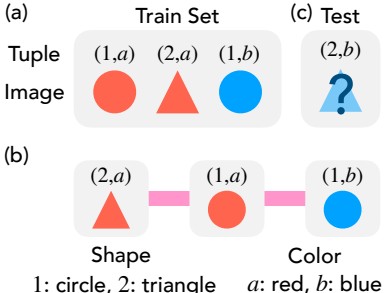

Figure 2: **Compositionality in a minimalistic conditional generation task.** (a) We train diffusion models on pairs of images and tuples, where the tuples denote the values of the *concepts* represented in the image (e.g., values of color and shape in the figure). (b) Samples between which only a single tuple element differs simplify the learning of a *capability* to recognize and alter the concept distinguishing the images. (c) To test the existence of such capabilities of a trained model, we ask the model to generate images corresponding to novel tuples that are out-of-distribution, hence requiring compositional generalization.

conditioning the model on tuples denoting which concepts an object in the image should possess, while systematically controlling the frequencies of concepts in the dataset. Thereafter, we study the model's ability to generate samples corresponding to novel combinations of concepts by conditioning the denoising process on a correspondingly novel tuple, thus assessing the model's ability to compositionally generalize. This approach allows us to systematically investigate the key properties of a dataset enabling compositional generalization in an interpretable and controlled manner in conditioned diffusion models. Our main contributions follow.

- *Concept graphs:* **A simple, synthetic framework for studying conditional diffusion models.** We develop a minimalistic abstraction for training conditional diffusion models by introducing a framework called "concept graphs". The framework forms the foundation of our systematic

study of how diffusion models learn to generate samples from a set of concepts and compose these concepts to generate out-of-distribution samples. To our knowledge, our results are the first demonstration of perfect compositional generalization on a (synthetic) vision problem.

- *Multiplicative emergence* **of compositional abilities.** We use our model system's interpretability to monitor learning dynamics of a diffusion model's capabilities and the ability to compose them to produce out-of-distribution data. As we show, a diffusion model first memorizes the training dataset and then sequentially generalizes to concepts that are at a greater "distance" from the training distribution. Since progress in learning each capability multiplicatively affects the model's performance in compositional generalization, we find a sudden ability to compose and produce samples out-of-distribution "emerges". We thus hypothesize compositionality may partially explain emergent phenomena seen in modern generative models [77–81].
- **Investigating challenges for compositional generalization.** We systematically examine the challenges arising from decreased frequency of specific concepts necessary for learning a corresponding capability. We further evaluate the effectiveness of using fine-tuning to address misgeneralizations in adversarial settings, and find that it is *generally insufficient* to enable the learning of new capabilities.

Before proceeding, we emphasize the trade-offs and limitations of our model-experimental systems approach. Just as neural mechanisms identified in model animals cannot be directly applied to human medical applications, our observations should not be considered definitive conclusions that can be directly transferred to modern large generative models. Instead, our study aims to establish conceptual frameworks, identify data-centric control variables, and formulate mechanistic hypotheses, paving the way for further theoretical and empirical explorations of larger models.

## 2 Related Work

**Diffusion models.** Diffusion models are the state-of-the-art method for generating realistic visual data [1–11]. They are often trained using image-text pairs, where the text is processed using a large language model to produce semantically rich embeddings that allow controlled generation of novel images and editing of existing images [26–31]. Such conditional diffusion models are easy to test for compositional generalization, as one can directly give a text description that requires composition of concepts the model is likely to have learned (e.g., avocado and chair) to produce images that were not encountered in the model's training data (e.g., avocado chair; see [1, 6]). Such results demonstrate the model's ability to compose and generalize out-of-distribution. However, we emphasize that text-conditioned models may fail to generalize compositionally if the text model is unable to sufficiently disentangle relevant concepts in the text-embedding space. To avoid this pitfall, we use ordered tuples that denote precisely which concepts are involved in an image's composition.

**Compositional generalization.** Compositionality is an inherent property of the real world [82], wherein some primitive such as color can be composed with another primitive such as shape to develop or reason about entirely novel concepts that were not previously witnessed [60]. Notably, compositionality is hypothesized to play an integral role in human cognition, enabling humans to operate in novel scenarios [83–87]. Inspired by this argument, several works have focused on developing [67, 68, 88–93] and benchmarking [64, 66, 90, 94–102] machine learning models to improve and analyze compositional generalization abilities. We especially highlight the works of Lewis et al. [96], who use a similar model experimental system approach as ours to evaluate factors influencing compositionality in contrastive language-image pretraining (CLIP) [34]. Our focus on generative models, instead of representation learning, distinguishes our results however. Relatedly, we note the notion of compositionality has been discussed in several recent works in the fields of disentangled and causal representation learning [101, 103–107]. Our framework for systematic understanding compositionality in generative models was heavily influenced by the synthetic dataset design considerations promoted in these papers. We also emphasize that a thorough formalization of compositionality in modern machine learning is generally lacking, but highlight the notable exceptions by Hupkes et al. [108] and Wiedemer et al. [106]. To avoid ambiguity, we provide a precise formalization that instantiates our intended meaning of compositionality.

## 3 Concept Graph: A Minimalistic Framework for Compositionality

In this section, we present the *concept graph* framework, as illustrated in Fig. 3, which enables us to visually depict compositional structure of our synthetic data and forms the basis for generating

hypotheses and designing experiments in our work. We begin by defining the essential building blocks of our framework: concept variables and concept values. In the following, we call a specific output of our data-generating process an "object" (see Fig. 3).

**Definition 1.** *(Concept Variables.)* *A concept variable $v_i$ uniquely characterizes a specific property of an object. $V = \{v_1, v_2, ..., v_n\}$ denotes a set of $n$ concept variables, such that, given an object $x$, $v_i(x)$ returns the value of the $i^{th}$ concept variable in that object.*

For instance, for geometric objects shown in Fig. 3, concept variables may include `shape`, `color`, `size`, `location`, and `angle`. These variables take on values from a pre-specified range, called concept values, as described below.

**Definition 2.** *(Concept Values.)* *For each concept variable $v_i \in V$, $C_i = \{c_{i1}, c_{i2}, ..., c_{ik_i}\}$ denotes the set of $k_i$ possible values that $v_i$ can take. Each element of $C_i$ is called a concept value.*

For example, in Fig. 3, the concept values for the `shape` variable are in the set $\{\text{circle}, \text{triangle}, \text{square}\}$, while for the `color` values are in the set $\{\text{red}, \text{blue}, \text{green}\}$. Given $n$ concept variables, with each variable $v_i$ having $k_i$ concept values, there can be $\prod_{i=1}^{n} k_i$ distinct combinations that yield a valid object. We use a unique combination of a subset of these concept variables, $v_1, \ldots, v_p$ $(p < n)$, to define the notion of a "concept class".

**Definition 3.** *(Concept Class.)* *A concept class $C$ is an ordered tuple $(v_1 = c_1, v_2 = c_2, ..., v_p = c_p)$, where each $c_i \in C_i$ is a concept value corresponding to the concept variable $v_i$. If an object $x$ belongs to concept class $C$, then $v_i(x) = c_i \,\forall i \in 1, \ldots, p$.*

Note that the remaining $n - p$ concept variables are free in the above definition. When these free variables are specified, we get a specific object from our data-generating process. That is, a concept class represents a family of objects by specifying the values of a pre-defined subset of concept variables. For example, in Fig. 1, different colored lizards instantiate images (objects) from the species (concept class) of lizards; here, color of the lizard serves as a free variable. Similarly, in the geometric objects scenario of Fig. 3, a "small red circle" would be a concept class for which the `shape`, `color`, and `size` variables have been assigned specific values. Objects are images designated by associating precise values with the remaining concept variables of `location` and `angle`. Next, we introduce the notion of concept distance, which serves as a proxy to succinctly describe the dissimilarity between two concept classes.

**Definition 4.** *(Concept Distance.)* *Given two concept classes $C^{(1)} = (c_1^{(1)}, c_2^{(1)}, ..., c_n^{(1)})$ and $C^{(2)} = (c_1^{(2)}, c_2^{(2)}, ..., c_n^{(2)})$, the concept distance $d(C^{(1)}, C^{(2)})$ is defined as the number of elements that differ between the two concept classes: $d(C^{(1)}, C^{(2)}) = \sum_{i=1}^{n} I(c_i^{(1)}, c_i^{(2)})$, where $I(c_{1i}, c_{2i}) = 1$ if $c_{1i} \neq c_{2i}$ and $I(c_{1i}, c_{2i}) = 0$ otherwise.*

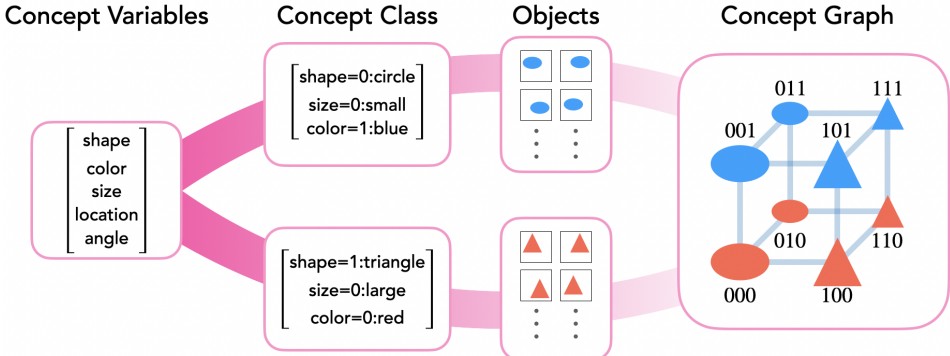

Figure 3: **Concept graphs.** We organize our study using a simple but expressive framework called *concept graphs*. The basis of a concept graph is a set of primitives called *concept variables* (e.g., `shape`, `color`, etc.). A subset of these variables is instantiated with specific values to yield a *concept class*, e.g., $\{\text{shape} = 0, \text{size} = 0, \text{color} = 1\}$ implies a small, blue circle. This is akin to defining a broad set of objects that share some common properties, e.g., all lizards in Fig. 1 belong to the same species, despite their diverse colors. A specific *object* in this class is instantiated by fixing the remaining variables, e.g., small, blue circles at different locations. Each concept class corresponds to a graph node, where nodes are connected if their concept classes differ by a *concept distance* of 1.

The concept distance quantifies the dissimilarity between two concept classes by counting the number of differing concept values. It is important to note that *this distance serves only as a null model* because each axis represents distinct concept variables and each of the variables can assume various possible concept values. We are now ready to define the notion of a concept graph (see Fig. 3), which provides a visual representation of the relationships among different concept classes.

**Definition 5.** *(Concept Graph.) A concept graph $G = (N, E)$ consists of nodes and edges, where each node $n \in N$ corresponds to a concept class, and an edge $e \in E$ connects two nodes $n_1$ and $n_2$ representing concept classes $C^{(1)}$ and $C^{(2)}$, respectively, if the concept distance between the two concept classes is 1, i.e., $d(C^{(1)}, C^{(2)}) = 1$.*

A concept graph organizes different concept classes as nodes in the graph, while edges denote pairs of concept classes that differ by a single concept value. An ideal conditional diffusion model, when trained on a subset of the nodes from this graph, should learn *capabilities* that allow it to produce objects from other concept classes. We formalize this as follows.

**Definition 6.** *(Capability and Compositionality.) Consider a diffusion model trained to generate samples from concept classes $\hat{C} = \{C_1, C_2, \ldots, C_T\}$. We define a **capability** as the ability to alter the value of a concept variable $v_i$ to a desired value $c_i$. We say the model **compositionally generalizes** if it can generate samples from a class $\widetilde{C}$ such that $d(\widetilde{C}, C_i) \geq 1 \forall C_i \in \hat{C}$.*

The ideas above are best explained via Fig. 4. Specifically, assume we train a diffusion model on data from a subset of concept classes, i.e., a subset of nodes in the concept graph. To fit the training data, the model might just memorize the training data or it *may* learn the relevant capabilities to alter specific concept variables; e.g., given samples from classes 0000, 0010, and 0001 in Fig. 4 (a), the model may just memorize the data or learn how to alter the third and fourth concept variables. Models that just memorize the training data lack the capability to generate samples from out-of-distribution classes (e.g., 1100). In contrast, the ability to compose concepts would enable the model to produce out-of-distribution samples starting from classes seen in the training data.

In summary, our concept graph framework provides a systematic approach to representing and understanding a minimalistic compositional structure, allowing for an analysis and comparison of different learning algorithms' abilities to generalize across various concept classes.

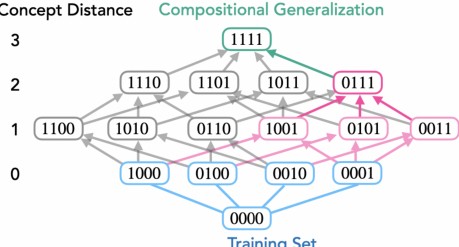

Figure 4: **Capabilities and compositionality in a concept graph.** Consider a lattice representation of a concept graph corresponding to four concept variables. Blue nodes denote classes represented in the training data; a model can either memorize data from these classes or learn capabilities to transform samples from one class to another. If it learns capabilities, it can compose them with data-generating process of samples from a concept class in the training data and produce samples that are entirely out-of-distribution, denoted as pink and green nodes.

## 3.1 Experimental and Evaluation Setup

With the concept graph framework in place, we now have the tools to systematically investigate the impact of different data-centric properties on learning and composing capabilities in conditional diffusion models. Before proceeding further, we briefly define our experimental setup and evaluation protocol. This allows us to interleave our results with our hypotheses, explaining them in context.

**Experimental Setup.** The detailed setup is presented in Appendix A. In brief, we train conditional diffusion models that follow the U-Net pipeline proposed by Dhariwal and Nichol [1]. Our dataset involves concept classes defined using three concept variables, each with two values; specifically, shape = {circle, triangle}, color = {red, blue}, and size = {large, small}. Tuples with binary elements are used to condition the diffusion model's training; they serve as an abstraction of text conditioning. For example, the tuple $000$ implies a large, red circle is present in the image. To sample images from this process, we simply map the size and color axes to the range $[0, 1]$ and uniformly sample to develop a training dataset of 5000 samples; the precise samples depend on which concept classes are allowed in the data-generating process. In this setup, the minimal required set for learning capabilities to alter concepts is four pairs of tuples

and images, each drawn from one of the following four concept classes: $\{000, (\text{circle}, \text{red}, \text{large})\}$, $\{100, (\text{triangle}, \text{red}, \text{large})\}$, $\{010, (\text{circle}, \text{blue}, \text{large})\}$, $\{001, (\text{circle}, \text{blue}, \text{small})\}$. Consider the following two tuples: $\{000, (\text{circle}, \text{red}, \text{large})\}$ and $\{100, (\text{triangle}, \text{red}, \text{large})\}$. In this case, we see that the shape concept is encoded in the first elements. Here, 0 represents a circle and 1 represents a triangle. The remaining concepts are similarly encoded in later elements.

**Evaluation Metric.** Evaluating whether a generated image corresponds to the desired concept class might require a human in the loop. To eliminate this requirement, we follow methods from literature on disentanglement and train classifier probes to predict whether a generated image possesses some property of interest [103, 109–114]. Specifically, we use the diffusion model training data to train three linear probes, which respectively infer the following three concept variables: `shape`, `color`, and `size`. We define a model's accuracy for generating images of a given concept class as the product of the probabilities outputted by the three probes that each concept variable matches the value for the concept class. Note that a random classifier for a concept variable will have an accuracy of 0.5. We indicate this random baseline with dotted, gray lines in our plots when useful.

## 4 Multiplicative Emergence of Compositional Abilities

We first investigate the order in which a model learn capabilities to produce sample from a concept class and how the learning of such capabilities affects the ability to compositionally generalize.

**Learning dynamics respect the structure of the concept graph (Fig. 5).** We find the ability to compositionally generalize, i.e., produce samples from out-of-distribution concept classes, emerges at a rate which is inversely related to a class's concept distance with respect to classes seen in training. Specifically, in Fig. 5 (a) we show the learning dynamics of the model, where lightblue nodes denote concept classes within the training dataset, while pink and darkpink nodes respectively denote classes at concept distances of 1 and 2 from classes in the training dataset. As the model learns to fit its training data (lightblue nodes), it learns capabilities that can be composed to produce samples from out-of-distribution concept classes (pink / darkpink nodes). Fig. 5 (b) further shows that the learning dynamics of compositional generalization are influenced by the concept distance from the training set: the model first learns the concept classes in the training dataset (lightblue lines) and then generalizes to concept classes with a concept distance of 1 from the training dataset (pink lines). Thereafter, the model suddenly acquires the capability to compositionally generalize to a concept class with a concept distance of 2 from the training dataset (darkpink line). Fig. 5 (c) shows the images generated by the model over time. We observe that rough shapes and sizes are learned relatively early in training, by the 4th epoch, while the color is highly biased to be red, the majority color in the training dataset, up to the 10th epoch. Then, around the 20th epoch, the model learns to generate the underrepresented

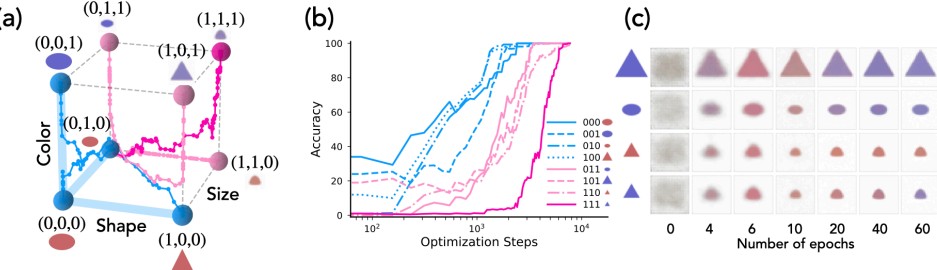

Figure 5: **Concept distance from the training set governs the order in which compositional capabilities emerge.** (a) Concept graph (cube) depicting training data points (blue nodes) and concept distances for test data points, where pink nodes represent distance = 1, and darkpink nodes represent distance = 2. Each trajectory represents the learning dynamics of generated images given a tuple prompt. Each trajectory represents the learning dynamics of generated images based on each tuple prompt. During every training epoch, 50 images are generated and binary classification is performed to predict each concept, including color, shape, and size. (b) Compositional generalization happens in sequence, starting with concept distance = 1 and progressing to concept distance = 2. The x-axis represents the number of epochs, and the y-axis represents the progress of compositional generalization. (c) Images generated as a function of time clearly show a sudden emergence of the capability to change the color of small, blue, triangles.

color (blue) for concept classes with a concept distance of 1. Finally, around the 40th epoch, the model learns to generate the underrepresented color (blue) for the class at distance 2, showing a sudden emergence of capability to generate samples from that class. The observations above generalize to higher-dimensional concept graphs as well (see Fig. 16). Specifically, we now introduce a fourth concept variable `backgroundcolor ={white, black}` and again find that compositional generalization occurs in the order governed by the distance from the training set: the accuracy begins to rise for concept distance of 1, continues to increase for distance 2, and peaks for 3.

**Multiplicative influence of capabilities drives the sudden emergence of compositional generalization (Fig. 6).** The interpretability of our experimental setup illustrates that the *multiplicative* reliance on underlying capabilities is the critical structure behind the sudden emergence of compositional generalization in our setup. For example, in Fig. 6 (a) we show the dynamics of an additive vs. multiplicative accuracy measure for generating concept class {111, (triangle, small, blue)}, which has a concept distance of 2 from the training data. We observe a sudden improvement of the multiplicative measure. To better understand the above, in Fig. 6 (b), we plot the accuracy of probes used for predicting each concept variable (shape, size, color). From the plot, we notice

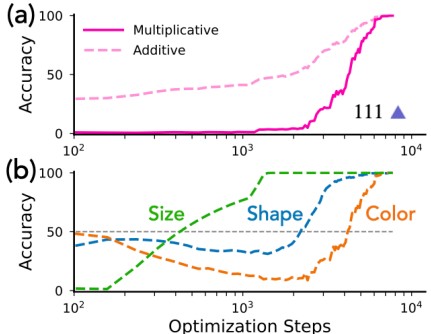

Figure 6: **Multiplicative influence of individual capabilities elicits an "emergence" of compositionality.** (a) Accuracy of producing samples from the concept class {111, (triangle, blue, small)}, which has a concept distance of 2 from the training data. A multiplicative measure (solid line) has a score of 1 when all concept variables of shape, color, and size are correctly predicted. Conversely, an additive measure (dashed line) independently relates each concept variable prediction accuracy to the compositional accuracy, deceptively suggesting smooth progress. (b) Learning dynamics of predicting each of the three concept variables: shape (blue), color (orange), and size (green).

that the model fails to acquire strong color transformation capabilities until the final stage of training, effectively bottlenecking compositional generalization of 111 class. This empirical observation leads us to the following hypothesis on *emergence of compositional abilities in generative models*.

**Hypothesis.** *(Compositional Abilities Emerge Multiplicatively.) We hypothesize the nonlinear increase in capability observed in large neural networks as size and computational power scale up is partially driven by the task's compositionality. Models must learn all required concepts, but compositional generalization is hindered by the multiplicative, not additive, impact of learning progress on each concept. This results in a rather sudden emergence of capabilities to produce or reason about out-of-distribution data.*

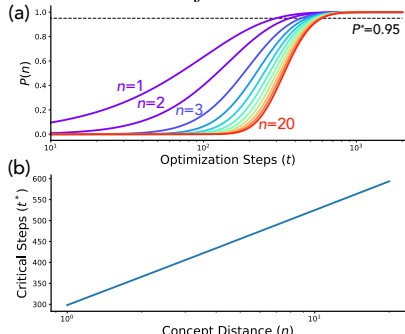

Figure 7: **Toy Model of Compositional Emergence.** (a) Probability of learning a compositional capability at a concept distance $n$ w.r.t. atomic abilities as a function of time $t$. (b) Critical optimization time steps $t^*$ required for the above probability to reach a threshold $P^*$ as a function of concept distance $n$.

The core aspect worth noting in our hypothesis is that compositional tasks require the concurrent acquisition of all involved "atomic" capabilities, akin to an AND logical condition. Correspondingly, in generative models, the learning of individual atomic abilities leads to a manifestation of emergent capabilities[1]. We argue this requirement leads to a non-linear increase in a model's capabilities (see also concurrent work making similar claims [78, 117]). We analyze this further via a simple toy model below.

**Toy analysis.** We consider a toy compositional problem which demonstrates how complex capabilities can rapidly develop ("emerge") from a set of atomic abilities. Assume there are $n$ atomic abilities, each with a probability $p$ of being learned in a given time step, i.e., the dynamics of learning an ability can be modeled as a Bernoulli coin flip: once the coin turns from 0 to 1, the model learns the said ability. The probability that the ability will be learned in $t$ steps is thus given by $1 - (1-p)^t$. An implicit assumption in this model is that the learning dynamics of

---

[1]This is related to the notion of "weak emergence" studied in psychology [115] and complex systems [116].

atomic abilities are independent (hence the name atomic) and that once an atomic ability is learned, it will not be forgotten. Now, our goal is to characterize the dynamics of learning a capability that involves composing these atomic abilities. The multiplicative emergence hypothesis argues that the learning such a capability by time $t$ requires that the model have learned the $n$ atomic abilities by that time. We then get the probability that the compositional capability has been learned by time $t$ is $P(n) = (1 - (1-p)^t)^n$. These dynamics are plotted in Fig. 7 (a) and look strikingly similar to the rapid learning (emergence) we found in our diffusion model setup (see Fig. 5 b)): for increasing degree of compositionality, we see much sharper learning dynamics (i.e., emergence) that is controlled by the concept distance w.r.t. atomic abilities. Further, assuming a given threshold probability $P^*$ at which one claims a capability has been learned, for a degree of composition $n$, we can solve for the critical time $t^*$ at which a compositional capability is learned as follows: $t^* = \left\lceil \frac{\log(1-(P^*)^{1/n})}{\log(1-p)} \right\rceil$. Plotting this in Fig. 7 (b), we see the progress on increasingly more compositional tasks is logarithmic w.r.t. the number of atomic concepts being composed (aka concept distance). Importantly, this implies if we allow the model to train infinitely and it learns several atomic abilities, it will have an explosion of capabilities due to the inherent compositionality of the data generating process—we further investigate this in a follow up work [118].

## 4.1 Challenges for Compositional Generalization

We have shown that, given a well-structured synthetic training dataset, a conditional diffusion model can learn to compose its capabilities to generate novel inputs not encountered in the training set. Next, we analyze an adversarial setups under which the model fails to learn relevant capabilities to compose concepts and if simple interventions can help mitigate these failures.

**Critical frequency for learning capabilities (Fig. 8).** We first probe the effect of changing the frequencies of samples from different concept classes in the training data and examine how this affects the model's ability to learn and compose capabilities involving that concept class. Results are shown in Fig. 8 and demonstrate how the frequency of color and size concepts in the training data impacts the generalization capabilities of the diffusion model. Specifically, we change the number of samples in the training data from 0 to 300 for concept class 001 (Fig. 8 (a)) and class 100 (Fig. 8 (b)). As can be seen, low frequencies of certain concept values degrade the accuracy of the model in both settings. Notably, training for out-of-distribution concept classes (pink lines) requires more samples than that for in-distribution ones (lightblue lines). This suggests that as the sample size grows, memorization occurs first, and generalization is achieved superlinearly as a function of data frequency. More importantly, we observe a critical number of samples are required before we can see the onset of capabilities to alter a concept. Specifically, in Fig. 8 (a), we can see that the model rapidly learns the color concept after being provided with 10 samples for a concept of large blue circle, 001. In contrast, in Fig. 8 (b), the model learns the shape concept only after reaching a certain threshold in the number of samples with a concept of large red triangle, 100.

We believe the results above are especially interesting because an often used strategy to prevent a generative model from learning harmful capabilities, such as the ability to generate images involving sensitive concepts, involves cleaning the dataset to remove images corresponding to such concepts, hopefully hindering the model's ability to generate samples containing the concept [33, 119, 120].

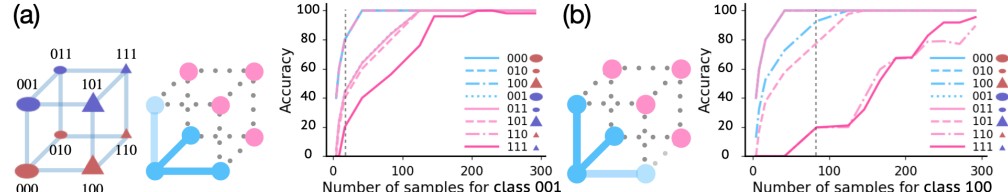

Figure 8: **How does the frequency of data samples impact the learning of capabilities?** We systematically control the frequency of a specific concept class in the training dataset and observe how it affects the model's learning of capabilities. (a) Capability to alter colors is quickly learned after introducing approx. 10 samples with a concept class of large blue circle, 001. (b) In contrast, a critical threshold (marked with dotted vertical lines) exists for learning a capability to alter the shape: as we gradually introduce samples, with a concept class of large red triangle, 100.

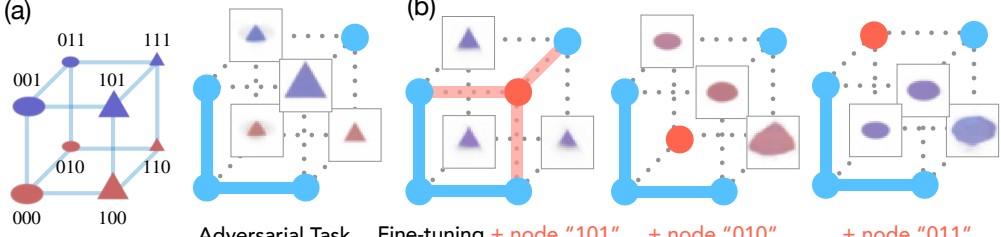

Figure 9: **Fine-tuning is not a remedy for misgeneralization in an adversarial task.** (a) The model struggles to generalize when the training setup is configured adversarially. As shown in the plot, the model incorrectly produces triangles (instead of circles) for both 011 and 010. (b) The model faces difficulty learning new concepts through fine-tuning. We add node 101 (triangle, large, blue) to the dataset and attempt to fine-tune the model. However, even with a large learning rate equal to the one used for training, the model fails to learn the capability to alter object size.

Such filtering is generally expensive and, arguably, statistically impossible to achieve perfectly in a large dataset, i.e., a few samples corresponding to the sensitive concept are likely to remain in the data. Our findings suggest that one might better consider the relationship between the probability of learning a concept and its frequency of occurence in the data to determine the degree of filtering required: if their presence in the training data is below a critical threshold, that can be sufficient to deter the model from learning a capability to generate samples related to that concept, obviating the need for a perfect filtering of the dataset.

**Fine-tuning is not a remedy for misgeneralization in an adversarial task. (Fig. 9).** We next evaluate a setting where concept classes present in the training data are not neighbors, i.e., their concept distances are greater than 1, but nonetheless represent all concept variables to allow a model to learn relevant capabilities. Specifically, as shown in Fig. 9 (a), we use only the following four pairs of tuples and images for training: $\{000, (\text{circle}, \text{red}, \text{large})\}$, $\{100, (\text{triangle}, \text{red}, \text{large})\}$, $\{001, (\text{circle}, \text{blue}, \text{small})\}$, $\{111, (\text{triangle}, \text{blue}, \text{small})\}$. Arguably, capabilities corresponding to shape change (000 to 100) and color change (000 to 001) can still be learned as the setup for these is similar to our prior experiments. However, our results show this yields an interesting failure mode; specifically, in Fig. 9 (a) we find the model fails to dissociate a second element value of 1 with the shape of a small triangle, and this bias causes the model to generate small triangles for both 011 and 010, when it should have produced small circles.

We further analyze if the failure above can be fixed via mere downstream fine-tuning (see Fig. 9 (b)). Specifically, we fine-tune the trained model on a dataset that includes the concept class of 101 (left), 010 (middle), and 011 (right). We find the model still generates small triangles for 011 and 010 even after fine-tuning. When the concept classes 010 (middle) and 011 (right) are added, the newly introduced concepts "overwrite" all existing concepts, causing the previously learned concepts (e.g., the color and shape concept for 101) to be forgotten. These results present a further challenge in addressing learned bias.

## 4.2 Additional Experimental Results with Real Data

To validate our hypotheses in more realistic settings, we conducted experiments using the CelebA dataset. We selected three attributes as the concepts: Gender (categorized as Female and "Male"), Smiling ("Smiling" and Not Smiling), and Hair Color ("Black Hair" and "Blonde Hair"). The concept graph corresponding to these attributes in the CelebA dataset is illustrated in Fig. 10 (a). For each individual concept, we trained a 3-layer CNN and adopted the product of accuracies from each concept as the evaluation metric. Although not all nodes achieved an accuracy of 100% due to limited training time, we find that our observations (partially) extend: (i) learning dynamics following the structure of the concept and (ii) delayed generalization of underrepresented attribute (gender), hold true, even at larger, more realistic scales. Fig. 10 (b) illustrates the learning dynamics using the real CelebA dataset. We again observe the sequential pattern in generalization. The accuracy begins to rise at concept distance 1 (denoted by pink lines), followed by concept distance 2 (red line). Notably, the node labeled 011—even though it is at concept distance 1—precedes the memorization stage (represented by blue lines). This exception can be attributed to the gender bias present in the training data. In Fig. 10 (c), we plotted the accuracy for the individual concept of gender (i.e., Female and Male). The Female concept class's training curve (red lines) reaches convergence faster

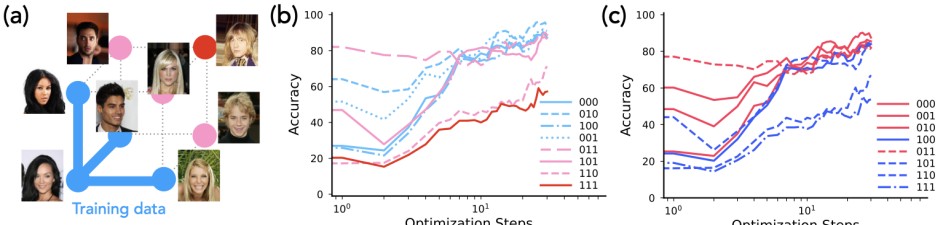

Figure 10: **Concept distance from the training set governs the order in which compositional capabilities emerge.** (a) Concept graph (cube) depicting training data points (blue nodes) and concept distances for test data points for CelebA dataset. (b) Compositional generalization happens in sequence, starting with concept distance 1 and progressing to concept distance 2. (c) Delayed emergence of abilities to generate underrepresented attribute (gender) for distant classes. Gender prediction accuracy based on generated samples at each step during training. We observe that the ability to generate underrepresented attributes occurs much later as concept distance increases.

compared to the Male concept class (blue lines). This disparity stems from the Female class being more predominant in the dataset, thereby making the Male class underrepresented. This observation offers useful insights for practitioners: to mitigate biased outcomes against underrepresented groups, we need to further train the diffusion model beyond its initial convergence on the training dataset.

## 5    Conclusion

In this work, we introduced a concept graph framework and used controlled interventions with a synthetic dataset to formulate and characterize four novel failure modes in the compositional generalization of diffusion models. We first hypothesized and verified that compositional generalization to a concept class far from the training data, based on concept distance, can fail, as it requires more optimization steps due to the sequential nature of generalization (Fig. 5). We show compositionality emerges in a sequence that respects the geometric structure of the concept graph, eliciting a *multiplicative emergence* effect that manifests as sudden increases in the model's performance to produce out-of-distribution data well after it has learned to produce samples from its training distribution. This behavior is reminiscent of the recently observed phenomenon of grokking in language modeling-like objectives [80, 81, 121]. Furthermore, when a particular concept value is underrepresented in the training dataset (e.g., an underrepresented color or gender), avoiding misgeneralization of the concept value, and thus achieving compositional generalization, requires far more additional optimization steps, even after achieving perfect performance on the training set. We also discovered a critical threshold for the inclusion of certain concepts in the training data, which is useful for deliberately making the model fail at learning harmful concepts (Fig. 8). Finally, we first analyze misgeneralizations in an adversarial task and find that fine-tuning does not provide a solution (Fig. 9). Overall, our synthetic data approach allowed us to apply controlled interventions that shed light on four different mechanisms behind the failure of diffusion models in compositional generalization.

## Authors' Contributions

ESL initiated the research by defining the problem. ESL and HT formulated the theoretical framework, the concept graph, and collaborated with MO on the experimental design. MO conducted most experiments and, along with ESL and HT, produced the visualizations and figures. ESL and HT authored the introduction. ESL was responsible for the literature review and related work. MO, ESL, and HT jointly worked on the results and discussion sections. Throughout the study, MO, ESL, and HT collaborated on analyzing experimental results and refining the manuscript. HT and RPD provided supervisory guidance and feedback.

## Acknowledgements

We thank Naomi Saphra, David Krueger and his group, Nikhil Vyas, Mohamed El Banani, and Anna Golubeva for fruitful discussions during the course of this project. ESL's time at University of Michigan was partially supported via NSF under award CNS-2008151.

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

# Supplementary Information

## A    Experimental details

### A.1    Synthetic dataset

The dataset consists of 5,000 rendered images of 2D geometric shapes, along with corresponding concept classes. These synthetic images are generated using Blender[2] by creating a scene graph and rendering it. Each scene contains a single object placed on a blank background of size $28 \times 28$. The objects have three types of attributes: size, color, and shape. There are two shapes (circle and triangle), three colors (red, blue), and two sizes (large, small), resulting in up to eight different combinations of attributes. Each image is annotated with the corresponding object attributes, which can be utilized as conditional features for image generation. This enables us to directly evaluate the text-to-image generation capability of the diffusion model against ground truth images. Fig. 11 depicts example images with the corresponding concept classes.

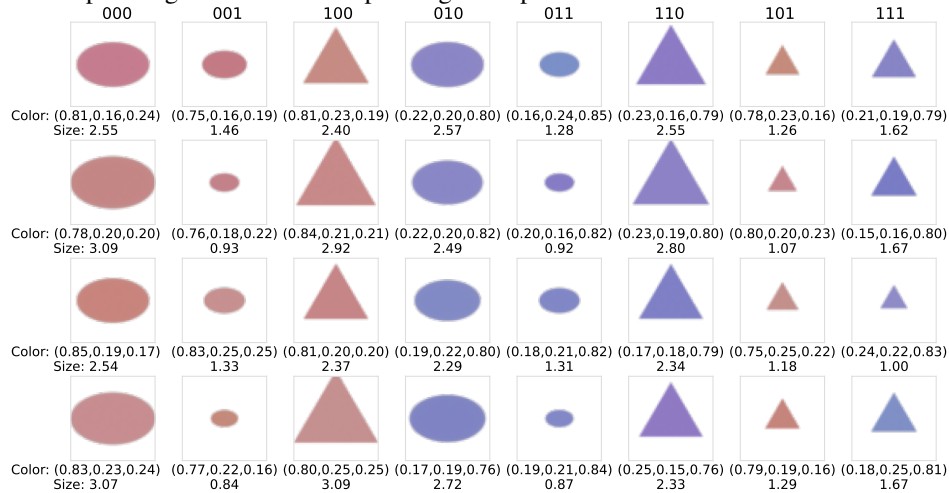

Figure 11: Examples of data samples with pairs of images and corresponding concept classes.

### A.2    Loss function

Diffusion models convert Gaussian noise into samples from a data distribution through an iterative denoising process. The sampling process starts with a noisy input $\mathbf{x}_T$, and denoised samples are generated through gradual iteration, $\mathbf{x}_{T-1}$, $\mathbf{x}_{T-2}$, until the original input $\mathbf{x}_0$ is obtained. Conditional diffusion models [10, 122] allow for a denoising process conditioned on texts or class labels. In all the experiments, we used the conditional diffusion model with the form $p(\mathbf{x}|V)$, where $\mathbf{x}$ denotes an image and $V = \{v_1, v_2, ..., v_n\}$ denotes a set of $n$ concept variables. To predict the noise $\epsilon$ at each timestep $t \in [0, T]$, we follow the approach proposed in [3] by training a neural network. Specifically, we construct a neural network $\epsilon_\theta(\mathbf{x}_t, t, \mathbf{a})$ and minimize the mean squared error (MSE) between the predicted Gaussian noise and the true noise:

$$\mathcal{L} = \mathbb{E}_{t \in [0,T], \mathbf{x}_0 \sim q(\mathbf{x}_0), \epsilon \sim \mathcal{N}(\mathbf{0}, \mathbf{I})} \left[ \|\epsilon - \epsilon_\theta(\mathbf{x}_0, t, V)\|^2 \right], \tag{1}$$

where $q(\mathbf{x}_0)$ denotes the distribution of input image $\mathbf{x}_0$, and $\mathcal{N}(\mathbf{0}, \mathbf{I})$ denotes the standard Gaussian distribution.

### A.3    Architecture

We use the conditional U-Net architecture [1], as in [3], for our neural network $\epsilon_\theta(\cdot)$. Our architecture comprises two down-sampling and up-sampling blocks, with each block consisting of $3 \times 3$ convolutional layers, GELU activation, the global attention, and pooling layers. The conditional information $V$ are fed through an embedding layer and concatenated with the image feature maps at each stage of the up-sampling blocks. We illustrate our network architecture in Fig. 12.

---

[2]http://www.blender.org

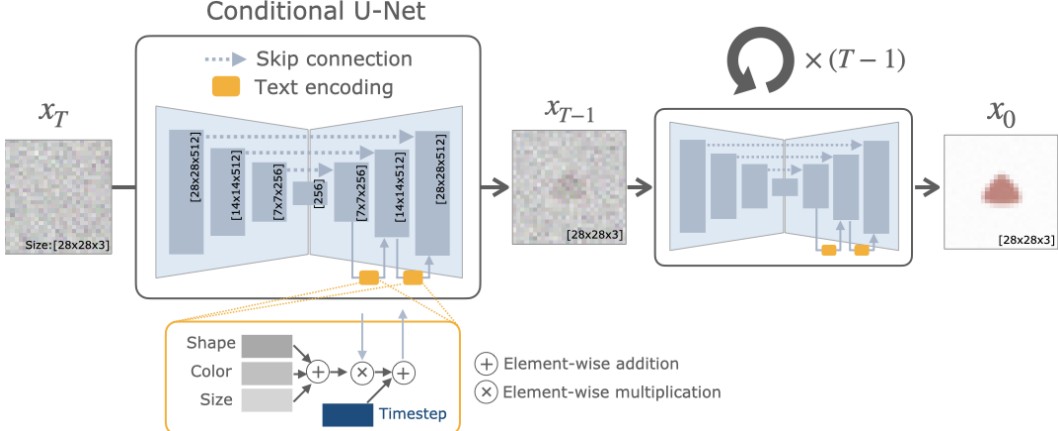

Figure 12: **The architecture of the conditional diffusion model.** The architecture of the conditional diffusion model involves an iterative process comprising noise addition and denoising steps. The model leverages conditioning information, specifically concept classes, to guide the transformation of the input image towards a desired state. In our implementation, we utilize a U-Net to parameterizethe denoising process. The U-Net architecture consists of three upsampling convolutional layers and three downsampling convolutional layers, which are connected through skip connections. Each layer within the U-Net includes a pooling layer, a global attention mechanism, and a GELU activation function.

## A.4 Optimizer

We implemented the diffusion model using PyTorch and trained it on four Nvidia A100 GPUs. We performed a hyperparameter search based on a validation set. We tested batch sizes ranging from 32 to 256, the number of channels in each layer from 64 to 512, leaning rate from $10^{-4}$ and $10^{-3}$, the number of steps in the diffusion process from 100 to 400. We employed the Adam optimizer [123] with $\beta_1 = 0.9$, $\beta_2 = 0.99$, and weight decay of $10^{-5}$.

## A.5 Evaluation metric

For the evaluation, we follow a probing protocol popularly used in several prior works on learning disentangled representations [103, 109–114]. To evaluate the attributes of the images generated by our conditional diffusion model, we trained linear classifiers on these images for three specific attributes: shape, color, and size. For each attribute, we developed a dedicated classifier: $f_0(\hat{x}_0)$ for shape, $f_1(\hat{x}_0)$ for color, and $f_2(\hat{x}_0)$ for size. Here $\hat{x}_0$ denotes the image generated by the conditional diffusion model. We utilized a cross-entropy loss function to train these classifiers. The output of each classifier fell into one of two categories: for shape, the categories were circle or triangle; for color, blue or red; and for size, large or small. We then calculated the accuracy for each attribute using the corresponding classifier outputs. To quantitatively assess the accuracy of predicted attributes aligning with their corresponding ground-truth concept classes, we utilize a multiplicative measure. This measure gauses the accuracy of all attributes, and defined by the product of individual accuracies for each attribute as follows:

$$\text{Accuracy} = \frac{1}{N_t} \sum_{n=1}^{N_t} \mathbb{1}\big(f_0(x_0^{(n)}), v_0^{(n)}\big) \cdot \mathbb{1}\big(f_1(x_0^{(n)}), v_1^{(n)}\big) \cdot \mathbb{1}\big(f_2(x_0^{(n)}), v_2^{(n)}\big), \qquad (2)$$

where $\mathbb{1}(\cdot)$ is the indicator function, $n$ is the index of the test samples, and $N_t$ is the total number of samples used for evaluation. For our experiments, we generated $N_t = 50$ images for each input of concept classes. $v_0^{(n)}$, $v_1^{(n)}$, and $v_2^{(n)}$ denote the actual (ground truth) concepts classes for shape, color, and size, respectively. We trained them over 50 epochs using the training dataset comprising 5,000 pairs of concept classes and images. The trained linear classifiers achieved an accuracy rate of 100% on the test set drawn from the original synthetic dataset.

Fig. 13, 14, and 15 display the average pixel values of the generated images with the prediction outcomes from the linear probing classifiers for the shape, color, and size attributes of these images, respectively. The figures in each subplot represent the ratio of accurately classified images out of a total of 50 samples. The predictions made by the linear classifiers align with our intuitive expectations, thereby confirming their suitability for evaluating the generated images.

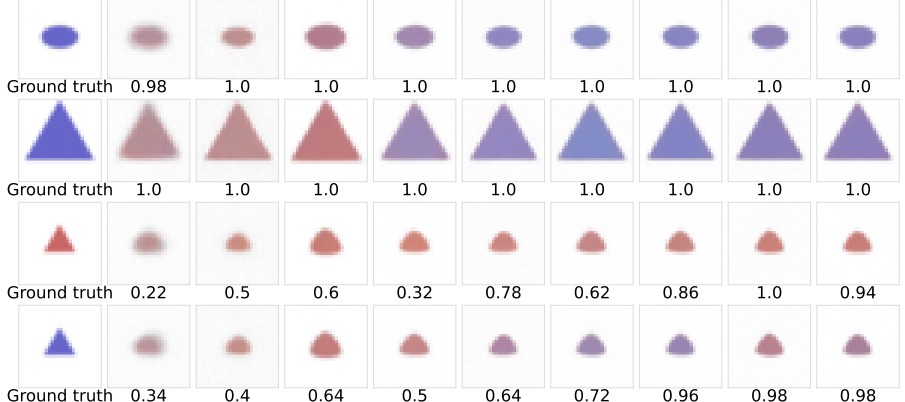

Figure 13: **Average pixel values of the generated images and prediction outcomes of the linear probing classifier for shape.**

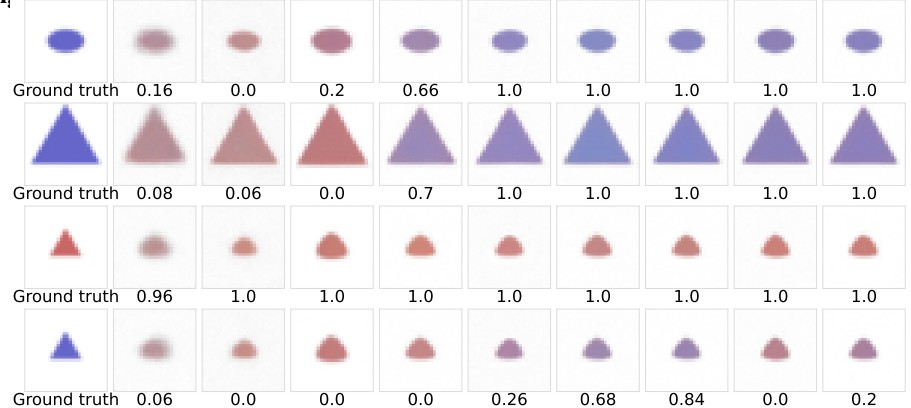

Figure 14: **Average pixel values of the generated images and prediction outcomes of the linear probing classifier for color.**

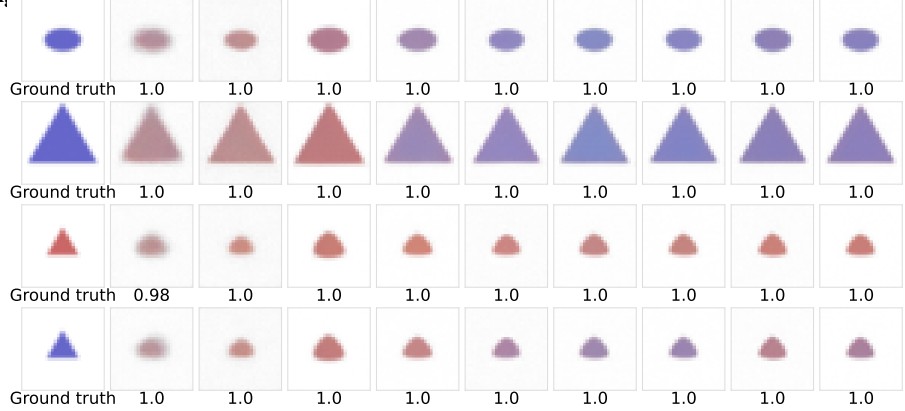

Figure 15: **Average pixel values of the generated images and prediction outcomes of the linear probing classifier for size.**

## A.6 Experiment with real data

We sourced our data from the CelebA dataset[3]. Our training dataset consists of 15,000 facial images, each labeled with its respective concept class. We delineated the concept classes based on these attributes: gender (classified as "Male" or otherwise), facial expression ("Smiling" or not), and hair color (either "Black Hair" or "Brown Hair"). We adjusted the images to a resolution of $48 \times 48$ pixels.

For our evaluation, we designed three classifiers for each concept using a three-layer CNN with 64 units. The learning rate was established at $10^{-3}$, trained the model for 10 epochs. We employed the ReLU activation function for the hidden layers and the sigmoid function for the output layer. We divided the image dataset into training and test sets at a ratio of 80% to 20%. The trained classifier exhibited a 96% accuracy for gender, 86% for facial expression, and 95% for hair color. All additional experimental configurations conformed to our synthetic data experiment procedures.

## B  Additional experimental results

In Fig. 16, we generalize the findings in Fig. 5 to a larger concept graph, i.e., one with four concept variables. Specifically, we now introduce a fourth concept variable backgroundcolor ={white, black} and again find that compositional generalization occurs in the order governed by the distance from the training set: the accuracy begins to rise at a concept distance of 1, continues to increase at a distance of 2, and peaks at 3.

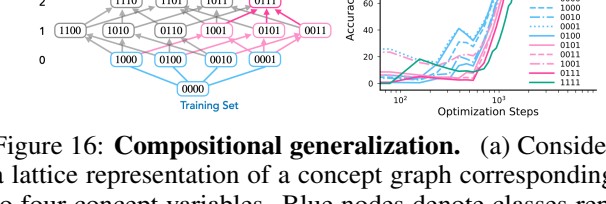

Figure 16: **Compositional generalization.** (a) Consider a lattice representation of a concept graph corresponding to four concept variables. Blue nodes denote classes represented in the training data; a model can either memorize data from these classes or learn capabilities to transform samples from one class to another. If it learns capabilities, it can compose them with data-generating process of samples from a concept class in the training data and produce samples that are entirely out-of-distribution, denoted as pink and green nodes. (b) We find compositional generalization happens in a sequence, starting with concept distance of 1 and then progressing to concept distance of 2 and 3.

In Fig. 17, we investigate the bottlenecks that affect compositional generalization by examining the prediction accuracy of concept variables such as shape, color, and size, as well as the overall classification accuracy. To evaluate this, we generated 50 samples in each epoch of training and classified them using a linear probe. Our findings reveal that various concept bottlenecks impede the learning process for different compositional objects. For instance, the generalization of blue-colored objects (011, 101, 111) is hindered by the color concept, indicating that the model struggles to correctly identify objects based on their color when it comes to these particular compositions. Similarly, the generalization to small red triangles (111) is bottlenecked by the shape concept, implying that the model faces difficulties in accurately recognizing the shape of these specific compositions. These observations highlight the challenges associated with compositional generalization and shed light on the specific concept bottlenecks that hinder the learning process. Understanding and addressing these bottlenecks can contribute to the development of more robust models capable of achieving better compositional generalization in various contexts.

In Fig. 18, 19, and 20 we present the experimental results for the diffusion model without global attention. Notably, we observe that these results exhibit a similar pattern to the ones obtained from the diffusion model with global attention, which is shown in Fig. 5 and 6. The experimental results consistently support our findings. Firstly, the order of emergence of compositional capabilities aligns with the compositional structure of the data-generating process, as evident in the results shown in Fig. 18. Secondly, we observe a notable delay in the emergence of the ability to generate underrepresented colors as the concept distance within a concept class increases. This emphasizes the importance of continued training beyond the point of achieving in-distribution generalization, as demonstrated in Fig. 19. Specifically, we observe that the model is capable of generating the majority color (red) much earlier than the underrepresented color (blue). Importantly, the training

---
[3]https://mmlab.ie.cuhk.edu.hk/projects/CelebA.html

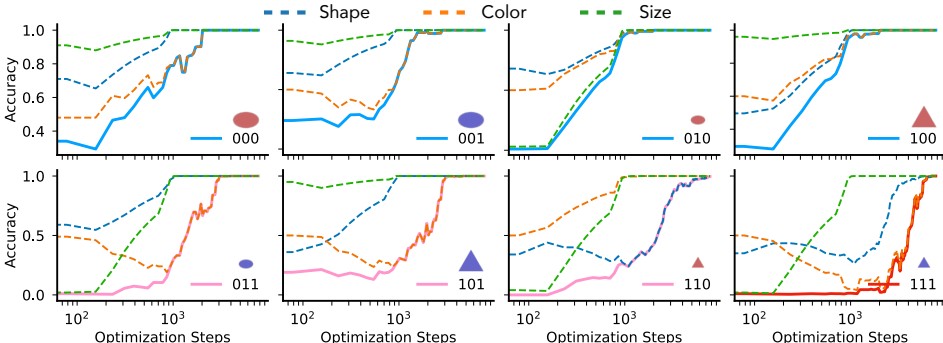

Figure 17: **Analyzing the bottlenecks of compositional generalization.** We examine the prediction accuracy of concept variables (shape, color, and size) and the overall classification accuracy. In each epoch of training, 50 samples were generated and classified using linear probe. We observe that various concept bottlenecks hinder the learning of different compositional objects. For example, compositional generalization to blue-colored objects (011, 101, 111) is bottlenecked by the color concept, while generalization to small red triangles (111) is bottlenecked by the shape concept.

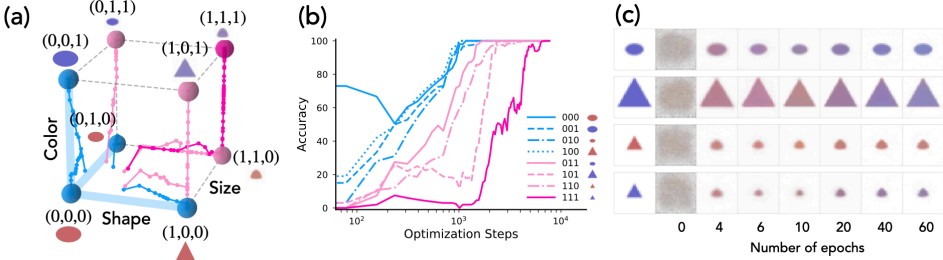

Figure 18: **Concept distance from the training set govern the order in which compositional capabilities emerge.** (a) Concept graph (cube) depicting training data points (blue nodes) and concept distances for test data points. (b) Compositional generalization happens in sequence, starting with concept distance = 1 and progressing to concept distance = 2. The x-axis represents the number of epochs, and the y-axis represents the progress of compositional generalization. (c) Images generated as a function of time clearly show a sudden emergence of capability to change color for small, red triangles.

time to achieve generalization based on a concept increases with concept distance. This observation provides important insights for training models with fairness in mind: even once generalization for in-distribution concept classes is achieved, stopping the training of a model will likely lead to a failure in generating underrepresented concepts, particularly for compositional generalization. Thirdly, our findings reveal a multiplicative impact of learning individual concepts on performance in compositional tasks, offering an explanation for the sudden emergence of capabilities, as depicted in Fig. 20.

In Fig. 21, we conducted experiments with different configurations by using varied sets of nodes in the training data. In Fig. 21(a) and Fig. 21(b), we inverted the axes of the concept graph from Fig. 9(a): (a) from bottom to top and (b) from left to right. In Fig. 21(c) and Fig. 21(d), we inverted the axes of the concept graph from Fig. 5(a): (c) from bottom to top and (d) from left to right. In Fig. 21(b) and Fig. 21(c), we observe that the model incorrectly generates triangles for both 011 and 010, even though circles are expected. In the other cases, the model outputs correct images for all the nodes. Our observations indicate that the model struggles to to dissociate a second element value of 1 with the shape of a small triangle, exhibiting a bias to generate triangles rather than circles for small objects.

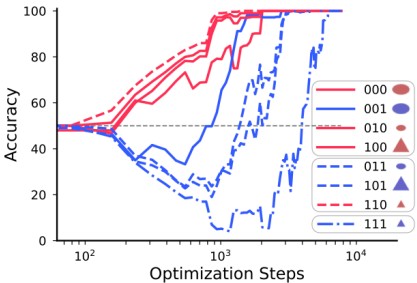

Figure 19: **Delayed emergence of abilities to generate underrepresented colors for distant classes.** Prediction accuracy of color based on generated samples at each epoch during training. We observe that the ability to generate underrepresented colors emerges significantly later as the concept distance of a concept class increases, highlighting the need for extended training beyond the point of achieving in-distribution generalization. This prolonged training enables effective composition of the underrepresented concept and leads to improved generalization.

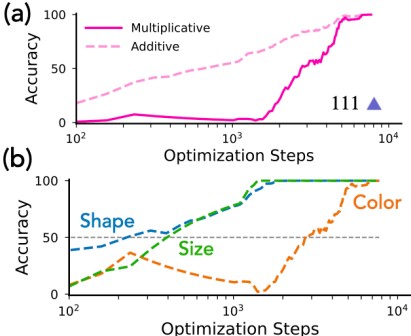

Figure 20: **Multiplicity underlies the sudden emergence of compositional capabilities.** (a) Accuracy of producing samples from the concept class {111, (triangle, blue, small)}, which has a concept distance of 2 from the training data. (b) Learning dynamics of accuracies for predicting each of the three concept variables: shape (blue), color (orange), and size (green).

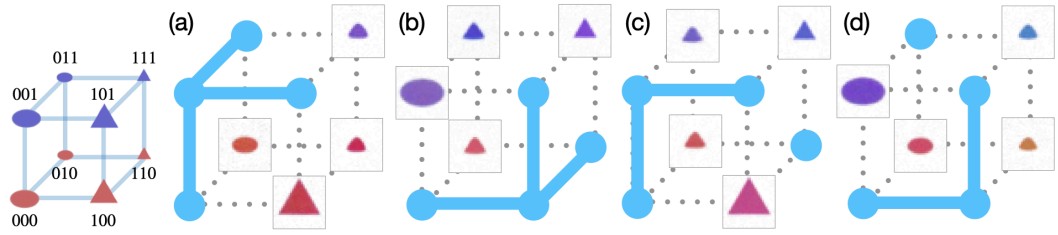

Figure 21: **The additional experimental results with different configurations by using varied sets of nodes in the training data.**