# OpenReview forum: "Compositional Abilities Emerge Multiplicatively: Exploring Diffusion Models on a Synthetic Task"
_NeurIPS.cc/2023/Conference — NeurIPS 2023 poster_

### Official Review · Reviewer_bYvo · 2023-06-15

**Soundness:** 3 good
**Presentation:** 3 good
**Contribution:** 3 good
**Rating:** 6
**Confidence:** 4

**Summary:**

The paper studies compositional generation abilities of conditional diffusion models. The main contributions of the paper are the concept graph framework, which is used to examine compositional abilities in a simplified setting, and insights on the learning dynamics of diffusion models.

**Strengths:**

The paper is very well written, motivated, and easy to follow. The problem the authors are investigating is relevant to know the limitations of compositional generation of the current diffusion models.

**Weaknesses:**

Even though the paper provides a very pleasing framework to investigate compositional capabilities of diffusion models through simple and highly controllable interventions, I think the submission is incomplete. This is mainly because the paper considers only a single architecture and does not provide guidelines how insights from concept graph framework could be further studied or provide validation that they hold in a large scale setting. Additionally, I would like to see a summary answering the questions from the abstract: “What are the reasons underlying this behavior? Which concepts does the model generally find difficult to compose to form novel data?”

**Questions:**

1. When does the sudden emergence of compositionality occur during training, can this be somehow predicted? Does it happen at the same time for different architectures and training configurations (noise schedules, augmentations etc.)? Providing answers to these questions would be valuable to researchers designing inductive biases to models.

2. How is the critical threshold for learning new capabilities in Fig. 8 determined? Is there a practical way of estimating critical thresholds for learning harmful concepts based only on the given training dataset?

3. Why Fig. 8 results could not be interpreted as if you have even a slight amount of harmful concepts in the dataset, then the model can eventually learn these given sufficient amount of training/capacity? From this point of view even a small amount of harmful samples leads to unwanted behavior and filtering training data is very important.

Minor points on improving the clarity:

4. Potential typo at line 188, “0010” corresponds to a sample in the training set.

5. Fig. 5 duplicate sentence. For clarity Fig. 5 b) and c) would benefit having the same units in the x-axis (either optimization steps or epochs).

6. I would remove the bullet point: “if so, under what circumstances does it fail?” from the beginning of Sec. 4 to Sec. 5 because that is where this question is assessed.

7.  Figure 8 lattices are potentially flipped. The lattice of (b) should be the lattice of (a) because in (a) the number of samples from “001” is modified, and vice versa. The lattice (b) is also redundant, I suggest refactoring the figure to contain three subfigures (a) showing the lattice and (b), (c) showing the interventions. Additionally, the x-axis shows the number of samples and the text discusses frequencies. I would switch the x-axis to show frequencies instead of the number of images to improve interpretability in a more general case.

8. Fig. 9 says “In this example, large objects are red, and small objects are blue.”, however in (a) there are large objects in both blue and red.
In general, figures would benefit if they were converted to vector graphics format (pdf, svg etc.) because they need to be zoomed in order to view.



**Limitations:**

The authors adequately addressed the limitations of their work.

---

> ### Author Rebuttal · Authors · 2023-08-10
>
> Dear Reviewer bYvo,
>
> Thank you so much for carefully recognizing the strengths of our work while providing us with concrete action items to make our submission more impactful and complete. In response, we have intensively experimented over the last week to generate three new plots for you (Figs. R2, R4, R5) and also drafted a new paragraph to summarize the results of our study. With these additions, we have fully addressed your concerns, and hope that you could now recommend this submission for acceptance.
>
> ---
>
> **Weaknesses:**
> - **Exploration of Architectures [Figure R4]:** Thank you for your great suggestion. In response, we conducted experiments with more diverse architectures and model configurations. Please refer to Figures R4 and R5, and the general response. We will include these new results in the final version of the paper.
> - **Lack of Guidelines on Concept Graph Framework [Figure R2]:** We agree that examining generalization to a large-scale setting would be a great way to demonstrate the robustness of our findings. In response, we have now run new experiments with more realistic data from CelebA to further verify our claims. Please refer to Figure R2 and the above general response for new experimental results and discussions. We confirmed that our findings on (i) learning dynamics following the structure of the concept graph and (ii) delayed generalization of minority attribute (gender), hold true, even at larger, more realistic scales. We will include these results in the final version of our paper.
>
> - **Summary Answering the Questions from the Abstract
> "What are the reasons underlying this behavior (unpredictable failure in compositional generalization)? Which concepts does the model generally find difficult to compose to form novel data?":**
> This is a great suggestion. In response, we have drafted a concluding paragraph to be included in the final draft:
> "*In this work, we introduced a concept graph framework and used controlled interventions with a synthetic dataset to formulate and characterize four novel failure modes in the compositional generalization of diffusion models. We first hypothesized and verified that compositional generalization to a concept class far from the training data, based on concept distance, can fail, as it requires more optimization steps due to the sequential nature of generalization (Figure 5). Furthermore, when a particular concept value is underrepresented in the training dataset (e.g., a minority color or gender), avoiding misgeneralization of the concept value, and thus achieving compositional generalization, requires far more additional optimization steps, even after achieving perfect performance on the training set (Figure 6). We also discovered a critical threshold for the inclusion of certain concepts in the training data, which is useful for deliberately making the model fail at learning harmful concepts (Figure 8). Finally, we found that correlations and biases in concept variables can make compositional generalization difficult, and fine-tuning does not provide a solution (Figure 9). Overall, our synthetic data approach allowed us to apply controlled interventions that shed light on four different mechanisms behind the failure of diffusion models in compositional generalization.*"
>
> ---
>
>
> **Questions:**
> - **Q1A Can we predict the sudden emergence of compositional abilities?**
> Great question! The key challenge in predicting the timing of the sudden emergence of compositional ability lies in tracking the progress during the plateau phase of learning dynamics, where the compositional loss remains flat. Our insight into multiplicative emergence suggests that we can break down the compositional task into individual tasks to make progress on each sub-task visible.
> - **Q1B Does the emergence of compositional ability happen at the same time for different architectures and training configurations (noise schedules, augmentations, etc.)?**
> As mentioned in the previous paragraph and in our general response, we conducted experiments with more diverse architectures and configurations. As you can see in Figures R4 and R5, the result shows rich phenomena, including how attention mechanisms accelerate compositional generalization.
> - **Q2 Response to how critical thresholds in Figure 8 are assigned and can they be practically estimated:**
> We compute the thresholds, represented by a dotted line, as the minimum quantity of samples required to achieve non-zero accuracy in distance 2 compositional generalization for “111.” However, making quantitative estimations of critical thresholds for learning harmful concepts solely based on the available training data may present challenges.
> - **Q3 Response to re-interpretation of results in Figure 8:**
> We share your perspective. To clarify, in lines 328–331, we emphasize that when the quantity of data falls below a certain critical threshold, additional data filtering becomes unnecessary. Another way to perceive this observation is that when the critical threshold is exceptionally low, a more rigorous data filtering approach becomes imperative. This alternative interpretation of our finding holds equal significance and we will expand the discussion in the paper to highlight the same.
>
> ---
>
> **Minor points:**
> Thank you so much for making all the suggestions for better presentation in minor points. We will reflect all of your feedback in the final draft.

---

> > ### Comment · Reviewer_bYvo · 2023-08-15
> >
> > Thank you for the interesting experiments and thorough answers to my feedback. The new data seems to support the claims of the paper.
> >
> > I acknowledge that testing your hypotheses using large scale real world data is challenging because controlling the attributes existing in the training data becomes more difficult. I found the experiment on CelebA an important step towards that direction and potential future work may extend your work to this direction. Additionally, the exploration of various diffusion architectures provide valuable insights of the learning dynamics and that the observations are not specific only to the initial model architecture selected in the paper.
> >
> > In my opinion, these experiments, rewriting the concluding paragraph, and improving the presentation quality of figures improve the submission significantly. I am happy to update my score accordingly.

---

### Official Review · Reviewer_wac1 · 2023-07-05

**Soundness:** 3 good
**Presentation:** 3 good
**Contribution:** 3 good
**Rating:** 5
**Confidence:** 4

**Summary:**

This paper proposes to empirically studied how compositional structure emerges in diffusion model. The paper proposes the abstraction of concept graphs, and illustrates how diffusion models first learn to fit the training data before compositionally generalizing. The paper illustrates how diffusion models have difficulty disentangling data and modeling low data concepts

**Strengths:**

- The paper is clearly written and the color coding through the text and figures greatly helped comprehension of the paper
- I enjoyed the introduction of concept graphs and the illustration of compositional distance with respect to graph
- The analysis in the paper is quite enlightening, illustrating how diffusion models first fit in distribution before generalizing out of distribution, as well as its struggles on finding confounding factors between variable and low data variables.

**Weaknesses:**

- The evaluation setting is rather simple and focuses on a single synthetic dataset with 3 classes of attributes
- It would be good to study generalization both to more complex images (for instance photorealistic synthetic scenes rendered using something like Kubric) as well as to more factors larger than 3
- In practice, a lot of concepts in the world follow the Zipf distribution -- it would be interesting to analyze that
- It would also be interesting to see how syntatic form would effect compositional generalization
- Some theoretical analysis on compositional generalization would also be interesting

**Questions:**

See above

**Limitations:**

Yes

---

> ### Author Rebuttal · Authors · 2023-08-10
>
> Dear Reviewer wac1,
>
> We sincerely appreciate the time and effort you invested in your thorough and insightful review of our submission. Your recognition of the paper's well-written aspects, alongside your positive remarks on the analysis and our approach to understanding diffusion models, has greatly encouraged us.
>
> We are equally grateful for your constructive suggestions to perform new experiments with (i) more than 3 attributes and (ii) more realistic images. Over the past week, we have dedicated ourselves to conducting those experiments and prepared two new plots (Figs. R1, R2) to directly address your feedback.
>
> With these improvements, we believe we have successfully addressed your concerns. We hope that you are satisfied with the revisions, and that this will strengthen your support for the acceptance of this submission.
>
> Below, we address your remaining questions and concerns in an effort to provide clarity on any remaining issues.
>
> **Simplicity of Evaluation Setting:**
> We emphasize that to test our hypotheses in a precise setup, we deliberately chose a simplified data-generating process. However, we agree that extending the results to more complex scenarios is valuable. In response, we have conducted additional experiments on a more complicated synthetic dataset with four attributes. Please refer to Figure R1 and the general response for the experimental results and observations; we found successful generalization of our claims in this setting as well. We will include these results in the revised manuscript to reflect your feedback.
>
> **Generalization to More Complex Images:**
> We agree that examining generalization to more complex and realistic images would be a great way to demonstrate the robustness of our findings. Accordingly, we have now run new experiments with more realistic data from CelebA to further verify our claims. Please refer to the above general response for the experimental results and observations. We confirmed that our findings on (i) learning dynamics following the structure of the concept and (ii) delayed generalization of minority color, hold true, even at larger, more realistic scales.
>
> **Analysis of Zipf Distribution and Role of Syntactic Form:**
> We find both these ideas to be very valuable and are keen to explore them further. Currently, we focus only on the goals of demonstrating that compositionality is indeed captured by modern generative models, that it likely underlies some emergent phenomena seen in their training, and the consequences of such sudden learning behaviors when biased datasets (i.e., correlated concepts) are used.
>
> **Theoretical Analysis on Compositional Generalization:**
> We emphasize that theoretical characterization of compositional generalization is extremely lacking in contemporary literature; in fact, there is little consensus on basic definitions of what compositionality means, as we note in the paper. By providing a formal framework and defining what we mean by compositionality, we do believe we have taken a step towards making a formal analysis amenable and are keen to follow up on this direction. Currently, we believe that pursuing this would divert attention from the primary motivation of the paper, and we therefore leave this for future work.

---

> > ### Comment · Reviewer_wac1 · 2023-08-21
> > **Rebuttal Response**
> >
> > I thank the reviewer for their rebuttal response -- I will maintain my current score as I think the paper would be strengthened much more if a different dataset other than CelebA is considered (for instance attributes of CLEVR or rendered Kubric datasets). The CelebA dataset is very biased and its difficult to access the accuracy in which concepts are correctly generated.

---

### Official Review · Reviewer_L63E · 2023-07-06

**Soundness:** 3 good
**Presentation:** 2 fair
**Contribution:** 3 good
**Rating:** 6
**Confidence:** 3

**Summary:**

The authors try to understand the compositionality aspects of generative models by training a conditional diffusion model in a toy setting on synthetic data. They show that the models indeed learn to be compositional if we train longer. They also hypothesize that the sudden emergence of compositionality in the later parts of training is due to multiplicity of how the model learns the underlying factors. They also show that it is hard to disentangle and generalize to new concepts if the underlying factors are highly correlated.

**Strengths:**

The paper is easy to follow. The study of compositionality in diffusion models is a crucial problem to understand. The authors studied it in a controlled setting which is novel.

**Weaknesses:**

The authors chose a very simple synthetic dataset to study this problem. The dataset has only 3 attributes. Even though some of the observations make sense, we have no guarantees that any of the results will extend to the real world.

The figures are quite confusing. Especially the colors. Blue (albeit a different shade), was used as an attribute and to denote the training data.

In Section 4, the authors used a bit of sensationalist language, such as “sudden emergence” etc, while such terms are not defined nor appropriately cited. They can improve the writing in this section.


**Questions:**

1.	Did you try or have results with synthetic data with > 3 attributes?
2.	Why did you not use synthetic data with real objects instead of colors and shapes? Verifying your claims on even a small real-object dataset would make the paper much more robust and useful.
3.	I might have missed this detail, what is the size of the test set, did you look at only the “pink” examples in the figures?

---

> ### Author Rebuttal · Authors · 2023-08-10
>
> Dear Reviewer L63E,
>
> Thank you for the insightful review of our paper. We are pleased that you found our research question crucial, our approach novel, and our writing easy to follow. Your specific and constructive recommendations to conduct additional experiments with more than three attributes and more realistic images were greatly appreciated. In the past week, we've committed to further experimentation and created two additional plots (Figs. R1, R2) in response to your feedback.
>
> We believe these improvements adequately address your concerns. We hope that you will find the revisions satisfactory and that this will solidify your support for the acceptance of our submission.
>
> **Question on Synthetic Dataset with >3 Attributes:** We emphasize that to test our hypotheses in a precise setup, we deliberately chose a simplified data-generating process. However, we agree that extending the results to more complex scenarios is valuable. We have accordingly conducted additional experiments with synthetic data with four attributes, adding “background color” as a new attribute, and have found our claims successfully generalize in this setup as well. Please refer to Figure R1 and the general response for the new experimental results and discussions. We will include these new results in the final version of the paper.
>
> **Question on Using Real Data:** We agree that examining generalization to more complex and realistic images would be a great way to demonstrate the robustness of our findings. Accordingly, we have now run new experiments with more realistic data from CelebA to further verify our claims. Please refer to Figure R2 and the above general response for new experimental results and discussions. We confirmed that our findings on (i) learning dynamics following the structure of the concept and (ii) delayed generalization of minority attribute (gender), hold true, even at larger, more realistic scales. We will include these results in the final version of our paper.
>
> **Question on the size of the test set and whether only “pink” nodes from concept graphs are evaluated:** As noted in Appendix A.5, we generate 50 novel samples via the diffusion model per target concept class, i.e., the class defined by combinations of different concept values, and report the accuracy of probes trained to predict whether a sample contains all underlying concept values that define it. Given 8 nodes, this yields an evaluation of 400 samples per concept graph. That is, the ability to generate samples from both in-distribution (shown as blue nodes in figures) and out-of-distribution (shown as pink nodes in figures) concept classes is evaluated.
>
> We also note that the “pink” nodes in the figures denote out-of-distribution concept classes, i.e., the model has never seen a sample that comes from that distribution during training. Evaluating the model’s ability to generate samples from such out-of-distribution classes allows us to show that a model can learn to produce samples from classes entirely unseen during training–a primary goal of our work. However, the evaluation does encompass the blue nodes as well (shown as blue lines in learning dynamics plots).
>
> **Clarification on Figures and Colors:**
> Thank you for this feedback! We will increase the diversity of colors used to represent different entities to reduce any possible confusion.
>
> **Concerning the use of the term “sudden emergence” in Section 4:**
> While we understand the criticism, we stress that the form of sudden generalization wherein a model does exhibit a sharp improvement in performance on a task (herein, generating samples from an out-of-distribution class) is unlike any standard notion of generalization we are taught in classical machine learning. Accordingly, we do firmly believe that a new technical term needs to be devised. To remain consistent with contemporary work [1], we currently use the term “emergence,” but are happy to use a more appropriate term if the reviewer has suggestions. We also emphasize that the use of “emergence” has a long history in related fields: e.g., in cognitive science, where emergence is often used to refer to sharp improvements in a child’s capabilities [2].
>
> [1] Wei et al., 2022. Emergent Abilities of Large Language Models.
>
> [2] Spelke, E.S. and Kinzler, K.D., 2007. Core knowledge. Developmental science.

---

> > ### Comment · Reviewer_L63E · 2023-08-18
> > **Thank you for the rebuttal**
> >
> > Thank you for the response. The rebuttal answered my questions. I am updating my score.

---

### Official Review · Reviewer_kruN · 2023-07-06

**Soundness:** 3 good
**Presentation:** 4 excellent
**Contribution:** 3 good
**Rating:** 7
**Confidence:** 5

**Summary:**

This work proposes a framework for studying the compositional generalisation abilities of diffusion models (or generative models more broadly). To that end it introduces the notion of a concept graph which the authors use to manipulate simple synthetic datasets. These concept graphs arrange different combinations of concept values and variables in a way that perseveres a notion of distance between different combinations (based on bit flips). By studying how performance changes as models are required to reconstruct concept value combinations that are further apart from the training data, they can draw inferences regarding the emergence of generalisation capabilities in diffusion models.

**Strengths:**

1. The article is very well written and easy to follow. The idea of the concept graph and the manipulation of the dataset is clearly explained and well motivated.
2. The use of a hypothesis driven research as opposed to the standard benchmarking approach is welcomed change compared to most of the research in AI. That the way the authors present it makes it seem as particularly novel is more a reflection on the field than this particular work (but see later on some suggestions regarding this point).
3. The point correlations between concept value make learning difficult for the models is important since it highlights how much model still rely on correlations in spite of the fact of their superior generative capabilities.

**Weaknesses:**

There are no substantial weaknesses, but there are some errors and omissions when discussing the literature which I will discuss below.


**Questions:**

1. Is Multiplicative Emergence unsurprising when the authors are forcing accuracy to be 0 if at least one concept value is incorrect during out-of-distribution generation?
2. I find it weird that colour takes longer to learn than the other properties. Since the reconstruction error must be very high even if the shape and scale are correct, I would expect cooler to be learned quicker. Do the authors have some insight into why this happens?
3. Is the relation to grokking really that strong? In grokking, models still exhibit high validation error even if the test error is high. But here

**Limitations:**

1. It is incorrect to characterise some of the work discussed in lines 120 and 121 as merely performing benchmarking. Indeed at least [94], [95] and [99] perform manipulations of the training and dataset in much the same way as this present work. Thus, while I definitely like that the authors took a more hypothesis driven approach to their work, it is not by any means the case that this is has not been done before. Other works that do this include [1] and [2]. While these may not be based on generative models, they are still relevant examples of the kind of study that the authors are advocating for.

2. Relatedly, the authors should discuss the relation of their work with [3] and [4]. Unlike the previous references, these do involve compositional generalisation capabilities of generative models. The former also provides one characterisation of compositional generalisation in generative models that is similar to the one used in this work. Thus it is indeed not the case that there have been no previous proposals and that [101] is the only noteworthy attempt (which by its own admission, just summarises previous definitions anyway).

3. The latter uses an approach based on transforming input images which allows it to identify the concepts that vary between images in a way that resembles the example presented in Figure 2, panel b. Their results show a similar pattern as the one in this work where combinations that are close to the training data are easier for the models. I believe that both this work and the previously discussed ones deserve a slightly deeper discussion.

4. Lines 291 and 292 are very close to each other which makes the caption of Figure 7 and the main text hard to distinguish.



[1] Hermann, K., Chen, T., & Kornblith, S. (2020). The origins and prevalence of texture bias in convolutional neural networks. Advances in Neural Information Processing Systems, 33, 19000-19015.

[2] Geirhos, R., Jacobsen, J. H., Michaelis, C., Zemel, R., Brendel, W., Bethge, M., & Wichmann, F. A. (2020). Shortcut learning in deep neural networks. Nature Machine Intelligence, 2(11), 665-673.

[3] Montero, M. L., Ludwig, C. J., Costa, R. P., Malhotra, G., & Bowers, J. (2020, October). The role of disentanglement in generalisation. In International Conference on Learning Representations.

[4] Montero, M., Bowers, J., Ponte Costa, R., Ludwig, C., & Malhotra, G. (2022). Lost in Latent Space: Examining failures of disentangled models at combinatorial generalisation. Advances in Neural Information Processing Systems, 35, 10136-10149.

** note that I used the same citation numbers the authors used for the other references i.e. [95] above corresponds to reference [95] in the main text.

---

> ### Author Rebuttal · Authors · 2023-08-10
>
> Dear Reviewer kruN:
>
> Thank you for your positive response! We are delighted that you find our paper "well written and easy to follow", the introduced ideas of the concept graph "clearly explained and well motivated", and the use of a hypothesis-driven approach "a welcome change to AI research." Below, we address your raised questions and comments.
>
> **Question on the role of the evaluation measure:** Great question! As we note in the appendix, defining a discrete measure that rewards correct concept values is intuitively more appropriate because, e.g., claiming a model can produce "avocado chairs" only makes sense if the generated image both looks like an avocado *and* a chair. However, to show our claims also hold for a more continuous measure, we have added a plot showing the progress of cross-entropy between ground-truth concept values and the predicted values via our probes (please see Figure R3). Even with this continuous measure, we observe the sudden emergence, as shown in Figure R3(a). This result is attributable to the multiplicative nature of our task setting, as highlighted in Figure R3(b).
>
> **Question on delayed learning of color:** Our hypothesis is that delayed learning of the capability to manipulate color is due to, in our setup, majority samples in a randomly sampled batch of data possess the same color, i.e., red. Consequently, the model quickly learns to produce the majority color, and then first learns to perform well on producing other concepts, before learning how to produce blue-colored samples, keeping the performance low on generating OOD concepts that require an ability to alter the color values.
>
> **Clarification on relation to grokking:** This is a fair point. The term "grokking" is generally supposed to refer to an inability to perform well on test/validation data well after the training error has reduced to zero. In our scenario, since the model is able to generate samples from within the train distribution, grokking is more "distributional," i.e., the model suddenly learns how to generate samples from unseen distributions. We will ensure this point is further clarified in the paper.
>
> ---
>
> **Limitations:**
>
> **Response to Comment 1:** We agree with the reviewer that papers like [94], [95], and [99] perform systematic manipulations of training data for evaluating the compositionality of neural networks. Due to space constraints, we were forced to use a reductive assessment of "benchmarking", but will add a much more detailed discussion to clarify this. For now, we note that our novelty lies in the use of a model-experimental systems approach for evaluating the compositionality of diffusion models, a thorough formalization of the notion of compositionality in this scenario, relating compositional behavior with emergent phenomena seen in modern generative models, and the consequences of these results on learning biases due to correlations in training data. We will also add discussions on the reviewer's cited works on disentanglement–we were well aware of these papers, but again space constraints disallowed a fruitful discussion of them.
>
> **Response to Comments 2 and 3:** Thank you for pointing out these references on disentanglement! While the goals of these papers are sufficiently different from ours (see the answer to L1 above), we will ensure a thorough discussion is included in the final version of the paper. We especially agree that the dataset structure in [3] is similar to ours, though our provided formalization and the precise results are very different.
>
> **Response to Comment 4:** Thank you for this feedback; we will ensure this is addressed in the final version of the paper.

---

> > ### Author Response · Authors · 2023-08-11
> > **Typo: Please replace "cross-entropy" with "probabilistic accuracy"**
> >
> > Dear Reviewer kruN,
> > We would like to clarify a typographical error in our earlier rebuttal. In our discussion pertaining to Figure R3, please replace "cross-entropy" with "probabilistic accuracy." Initially, we experimented with cross-entropy loss, but for consistency with the notion of accuracy, we transitioned to this updated measure. We apologize for any confusion and direct you to our general response for further details.

---

> > ### Comment · Reviewer_kruN · 2023-08-14
> >
> > I agree with the issue of novelty when compared to previous work and look forward to the authors updated discussion of previous work and its relation to their own work which I do agree is good addition.
> >
> > Thanks for the reply and will certainly update my score accordingly.

---

### Author Rebuttal · Authors · 2023-08-10

Dear Reviewers,

We thank all reviewers for their diligent efforts in evaluating our submission. We are pleased by the unanimous recognition and support for our scientific approach, aimed at enhancing the understanding of diffusion models using minimal synthetic data. We would also like to thank the reviewers for the constructive and actionable feedback to further strengthen our claims in more diverse and realistic setups. In the attached one-page PDF, we have incorporated **five** new experimental figures that cover all the requested additions. With these enhancements, we are now confident that our paper will constitute a unique and impactful contribution to the NeurIPS conference.

---

**Summary of new experimental figures in response to reviewers' feedback:**

*   **[Figure R1] Sequential Generalization on Concept Graph Holds with 4 (>3) Attributes (L63E, wac1):** We have performed new experiments with 4 attributes. Here, we introduced a new concept of "background color" (ranging from 0 for black to 1 for white) in addition to the three existing concepts of "shape," "object color," and "size." Through this experiment, we confirmed that our hypothesis of “Sequential compositional generalization on concept graphs” generalizes to the case with more than three attributes. Memorization occurs first, as represented by the blue lines. Subsequently, generalization happens in sequence based on the concept distance: Concept distance 1 (depicted by pink lines) comes first, followed by concept distance 2 (red lines). Lastly, concept distance 3, represented by the green line, arises. Overall, our claim holds more generally beyond the concept cube!

*   **[Figure R2] Our Observations Hold in More Realistic Dataset (bYvo, L63E, wac1):** To generalize our hypotheses in more realistic settings, we have performed experiments with the CelebA dataset. We have chosen three attributes for our concepts: Gender (categorized as Female and "Male"), Smiling ("Smiling" and Not Smiling), and Hair Color ("Black Hair" and "Blonde Hair"). The corresponding concept graph for these attributes in the CelebA dataset can be found in Figure R2(a). We trained a 3-layer CNN for each individual concept and used the product of accuracies from each concept as the evaluation measure. While not all nodes reached an accuracy of 100% due to insufficient training time, our results consistently support our primary observations:

      *   **Learning dynamics respect the structure of the concept:** Figure R2(b) illustrates the learning dynamics using the real CelebA dataset. We again observe the sequential pattern in generalization. The accuracy begins to rise at concept distance 1 (denoted by pink lines), followed by concept distance 2 (red line). Notably, the node labeled '011' – even though it's at concept distance 1 – precedes the memorization stage (represented by blue lines). This exception can be attributed to the gender bias present in the training data, which we'll explain below.
      *   **Delayed Generalization of Minority Attribute:** In Figure R2(c), we plotted the accuracy for the individual concept of gender (i.e., Female and Male). The Female concept class's training curve (represented by red lines) reaches convergence faster compared to the Male concept class (depicted by blue lines). This disparity stems from the Female class being more predominant in the dataset, thereby making the Male class a minority. This observation offers useful insights for practitioners: To mitigate biased outcomes against minority groups, we need to further train the diffusion model beyond its initial convergence on the training dataset.

*   **[Figure R3] Emergence Persists with Probabilistic Accuracy Measure (kruN):** In Figure R3, we plot the learning dynamics for the probabilistic accuracy. Here, the probabilistic accuracy is calculated as the product of probabilistic accuracies $p(f(x^{(n)}), v^{(n)})$ from each individual concept, where $n$ is the index of the test sample, $f(\cdot)$ represents the classifier, $x^{(n)}$ is the generated image, and $v^{(n)}$ denotes the actual (ground truth) concept class (i.e., attribute). Using this probabilistic metric, we once again observe the sudden emergence of capability in the test nodes (represented by pink and red lines), especially for node "111" (highlighted by the red line). This confirms that our observation of the emergent capability isn't solely due to the choice of evaluation metric.

*   **[Figure R4] Attention Mechanisms Accelerate Compositional Generalization (bYvo):** In Figures R4(a)-(c), we explored three distinct architectures as the backbone of the diffusion model: U-Net without global attention (Figure R4(a)), U-Net with global attention (Figure R4(b)), and the Transformer model (Figure R4(c)). Our observations indicate that the models with attention mechanisms initially exhibit a slower learning curve. However, once they commence learning, there is a notable surge in their accuracy. Specifically, in Figure R4(c), the Transformer's accuracy initially increased slowly but quickly peaked at 100% shortly after.
*   **[Figure R5] Our Observations Hold Robustly across Diverse Configurations (bYvo):** Figures R5(a)-(c) show the learning dynamics across different hyperparameter settings for the diffusion model. Specifically, we varied the number of units in U-Net, labeled as "units" in the figures, and the number of time steps in the diffusion model, denoted as "time steps" in the figures. We verified that our findings regarding (i) sequential generalization of capability based on concept distance and (ii) sudden emergence of capability remain consistent across all hyperparameter settings. Based on your insightful recommendations, we are carrying out extensive experiments, e.g., using various noise schedules. However, due to page limitations, we have omitted some of these. Importantly, all the results we have now support our primary observations!

---

> ### Comment · Reviewer_kruN · 2023-08-14
>
> I would like to thank the authors for the detailed response and the additional experiments. I will respond here to their general comments and to their specific response to my original comment over there.
>
> First, I would like some further clarification for Figure R4. It seems from the plot that the advantage in performance lies with the U-Net with Global Attention, not the Transformer. In fact it's not clear to me (at least from this plot) that the Transformer has a substantial advantage over the vanilla U-Net outside of the the training data (it does train faster so I guess there is that). Maybe a better way to plot this is to have one plot per condition (blue, pink and red) and for each of these plot the three models together. For a final version you may need to run more seeds if the results do not show a clear effect (which can admittedly be a bit subjective even if using things like t-tests and the like).
>
> I like the plots in Figure R3, but the caption needs be clearer, or in the case of the bottom panel of b) actually there, as it doesn't even say what those three colours stand for. While I agree that the measure you propose is more consistent with accuracy, I would still like to see the plots with cross entropy. My insistence is based on an analogous reasoning found in [1]. Slightly different topic, but I think the reasoning applies here as well.
>
> [1] Schaeffer, R., Miranda, B., & Koyejo, S. (2023). Are emergent abilities of Large Language Models a mirage?. arXiv preprint arXiv:2304.15004.

---

> > ### Author Response · Authors · 2023-08-14
> >
> > Thank you for your response and updating your score!
> >
> > **Regarding clarification for Figure R4:** Due to short time window for rebuttals, we primarily focused on confirming our results generalize to a different architecture, i.e., Transformer for now. We agree with your assessment that the Transformer's primary advantage seems to be its faster training speed, but we will ensure to exhaustively probe this assessment further before making a general claim in the paper. Specifically, we are planning to conduct the following experiments:
> > - Additional experiments with varied hyperparameter settings and seeds, and include the results in the final version.
> > - Make individual plots for each condition, showing the performance of the different models.
> > - Add the discussion in our manuscript to provide a clearer explanation of the comparative performances.
> > - Explore the use of statistical tests.
> >
> > **Regarding the caption for Figure R3:** We appreciate your feedback on this matter. NeurIPS rules say our 1-page pdf shouldn't have much text, so we kept the captions brief. In Figure R3(b), the green line denotes the concept of size; the blue line denotes shape; and the orange line denotes color. We will add a clearer caption in the final version.
> >
> > **Cross-entropy plot:**
> > Indeed, we originally plotted the cross-entropy loss and observed the sudden emergence in it. We can’t modify the figures at this stage, but we are certainly happy to include the cross-entropy plot and discuss its relationship to reference [1] in the final version as well!

---

### Decision · Program_Chairs · 2023-09-21

**Decision:**

Accept (poster)

**Comment:**

This work studies compositional generalization of conditional diffusion models. All four reviewers are positive about this work. The compositionality of diffusion models is crucial and the controlled study is novel, the idea of concept graphs is interesting, and the finding on learning dynamics is intriguing that diffusion models first fit in distribution before generalizing out of distribution. Main concerns are lack of theoretical justification and whether the results and findings extend to large scale realistic world. AC agrees this work sheds new insights into the understanding of diffusion models, and thus recommends acceptance.